# Nanocarriers for Delivery of Oligonucleotides to the CNS

**DOI:** 10.3390/ijms23020760

**Published:** 2022-01-11

**Authors:** David Male, Radka Gromnicova

**Affiliations:** Department of Life, Health and Chemical Sciences, The Open University, Milton Keynes MK7 6AA, UK; Radka.Gromnicova@Open.ac.uk

**Keywords:** nanocarrier, oligonucleotide, blood-brain barrier, brain endothelium, gold nanoparticle, CNS diseases

## Abstract

Nanoparticles with oligonucleotides bound to the outside or incorporated into the matrix can be used for gene editing or to modulate gene expression in the CNS. These nanocarriers are usually optimised for transfection of neurons or glia. They can also facilitate transcytosis across the brain endothelium to circumvent the blood-brain barrier. This review examines the different formulations of nanocarriers and their oligonucleotide cargoes, in relation to their ability to enter the brain and modulate gene expression or disease. The size of the nanocarrier is critical in determining the rate of clearance from the plasma as well as the intracellular routes of endothelial transcytosis. The surface charge is important in determining how it interacts with the endothelium and the target cell. The structure of the oligonucleotide affects its stability and rate of degradation, while the chemical formulation of the nanocarrier primarily controls the location and rate of cargo release. Due to the major anatomical differences between humans and animal models of disease, successful gene therapy with oligonucleotides in humans has required intrathecal injection. In animal models, some progress has been made with intraventricular or intravenous injection of oligonucleotides on nanocarriers. However, getting significant amounts of nanocarriers across the blood-brain barrier in humans will likely require targeting endothelial solute carriers or vesicular transport systems.

## 1. Introduction

The development of gene therapy has opened up prospects for the treatment of diseases that affect the central nervous system [1,2]. Many monogenic diseases, particularly enzyme deficiencies, have their primary pathology in the CNS. These conditions are important candidates for gene therapy, by correction or replacement of the defective gene. Other conditions affecting the CNS, such as neurodegenerative diseases, can be alleviated by altering individual gene expression, even when the underlying pathology is complex or polygenic [3].

Several approaches are available for the treatment of monogenic diseases. The complete replacement of a defective gene involves delivery of a large segment of double-stranded genomic DNA (5–10 kBa) to target cells in the CNS. The gene may be integrated into the host cell DNA or act as an episome. Viral vectors are most suitable for delivery of entire genes to cells of the CNS, and this approach has achieved some success particularly with adeno-associated viruses (AAVs), which can hold up to 5 kBa DNA [4]. Nucleic acids, particularly RNA, are susceptible to nucleases in serum and tissue fluids and the viral vectors protect the DNA/RNA within the viral capsid. The first-generation vectors were mostly optimised for transfection of cells, gene integration and expression, rather than their ability to cross the blood-brain barrier. For this reason, delivery of the viral vectors has often been by direct intrathecal or intraventricular injection. The great majority of research and clinical trials on CNS-gene therapy has been with viral vectors, and this area has been well reviewed [2,5]. Viral vectors have many advantages for delivery of large gene segments or entire genes. Recent developments in gene therapy have focussed on the use of oligonucleotides [6,7,8]. Viral vectors have also been used to transfect cells with these short gene-segments, which can modify the activity of defective genes. However, viral vectors do induce neutralising antibodies or cytotoxic T cells, which can limit their effectiveness even in individuals who have not previously been treated [9].

A number of non-viral delivery systems have also been investigated. Initially, these methods were based on biochemical reagents and intended to produce maximum transfection of the target cells [10]. The use of nanoparticle carriers (nanocarriers) for oligonucleotides, which is a more recent development, is the subject of this review.

Nanocarriers are well suited to the transport of oligonucleotides, although they have also been used for entire genes. The requirements for such a nanocarrier can be summarised as:Protect the nucleic acid against enzymatic degradation,Non-toxic and low immunogenicity,Selective internalisation into the target cells of the CNS,Release of the nucleic acid and appropriate expression or activity in that cell type.

These considerations apply to any nanocarrier that is directly injected into the CNS. However, nanocarriers injected intrathecally are limited to the site of injection and if injected into ventricles, they preferentially distribute to areas around the ventricles [11]. Ideally the nanocarriers would be given intravenously, to achieve widespread distribution, but this means that they must cross the blood-brain barrier, which presents three additional requirements:The nanocarriers must not be cleared through the kidney, or removed by mononuclear phagocytes in the liver and spleen, before they have had time to interact with the brain endothelium,They should selectively target the brain endothelium, in comparison with endothelium in other tissues,They must cross the brain endothelial cells with their nucleic acid cargo intact and in sufficient quantity to have a biological effect.

To produce a nanocarrier that fulfils all of these requirements is extremely challenging [12]. Progress has been made in each of these areas, but no formulation can yet address all of the requirements.

## 2. Barriers for Nanoparticle Transport into the CNS

The blood-brain barrier presents both an anatomical and biochemical barrier for movement of large biomolecules from the capillaries to the brain parenchyma [13,14]. Anatomically, it is formed by the brain endothelium, astrocyte foot processes and the basal lamina—together, the endothelium and astrocytes form an apposed double basal lamina that contains pericytes. However, it is primarily the endothelium that prevents or facilitates movement of nanoparticles. Potentially, there are three routes by which nanoparticles could cross the endothelial barrier.
Paracellular movement through the junctions between endothelial cells.Direct movement across the apical plasma membrane, cytosolic transport or diffusion across the cell and transfer across the basal plasma membrane.Vesicular transcytosis—endocytosis at the apical (blood) surface of the endothelium and exocytosis at the basal (brain) surface.

Each of these routes is highly restrictive. The continuous, tight junctions between brain endothelial cells are formed by fusion of the outer membrane leaflets of adjacent cells by occludin and members of the claudin family including claudin-5. The multiple strands of tight junction proteins extend over 200–500 nm and they prevent paracellular diffusion of the great majority of polar molecules >1 kDa molecular weight. The smallest nanocarriers are 2–3 nm in diameter and 20–30 kDa—a similar size to plasma proteins—and they are unable to move through the intercellular junctions. In some circumstances, the junctions can become leaky—for example, at sites of inflammation—in response to inflammatory cytokines and chemokines (TNFα, IFNγ, CCL2, etc.) [15]. Viral infection, hypoxia and some toxins also weaken junctions. Such changes could allow nanoparticles to selectively enter the CNS at areas affected by inflammation or damage. However, the barrier is still substantially maintained even in areas of severe inflammation. The new capillaries in some glioblastomas also lack a barrier or have reduced barrier properties. The effect is quite variable between tumours, and this characteristic has been used to promote delivery of cytotoxic drugs to tumours that would otherwise be blocked by ABC-transporters in the endothelial plasma membrane. This feature can facilitate delivery of small cytotoxic drugs. However, unless the tight junctions are completely disrupted, there is little prospect for movement of any type of nanoparticle into the CNS by the paracellular route.

Some very small nanoparticles can directly cross the plasma membrane by a mechanism called ‘snorkelling’. This involves temporary integration into the plasma membrane and reorganisation of the surface ligands on the nanoparticle [16]. This effect has been seen with gold glyconanoparticles (<5 nm) and it results in transfer to the cytosol. Movement to the brain requires a similar trans-membrane transfer at the basal surface. While this process can be demonstrated in vitro [17], it is doubtful whether a sufficient number of nanoparticles could be transferred by this route and carry sufficient nucleotides to have a significant biological effect in vivo.

Vesicular transcytosis is potentially the best route for bulk transfer of large molecules or nanocarriers across the brain endothelium (Figure 1). Caveolae are characteristic endothelial transport vesicles. They are typically 50–100 nm in diameter and can, therefore, accommodate relatively large nanoparticles. The normal brain endothelium has relatively few caveolae in comparison with other endothelia, but this increases in areas of inflammation and damage.

Transport vesicles are generally selective in terms of the material they transport. Specific transport can be absorptive or receptor-mediated. In absorptive transport, the cargo interacts with the endothelial glycocalyx or surface glycoproteins at specialised zones of the plasma membrane called lipid rafts. In the brain endothelium, the glycocalyx has a particularly high negative charge due to sulphation of carbohydrates, and this promotes binding of positively charged serum molecules or nanoparticles [18].

In receptor-mediated transport, the specific ligand binds to the receptor on the endothelial surface before internalisation. Many solute carriers are present on the brain endothelium for transport of small metabolites required by the CNS. The solute carriers generally transport small molecules across the apical plasma membrane to the cytosol, and therefore, are not suitable for direct transport of nanocarriers. However, the solute carriers can be used for selective targeting of nanoparticles to the brain endothelium. More relevant for vesicular transcytosis are the receptors that transport large biomolecules to the CNS, including the transferrin-receptor (TfR1) LDL-receptor (LRP1) and insulin receptor. The targeting of nanocarriers is outlined in more detail in Section 4.3.

In some areas of the brain, the endothelium is fenestrated and nanoparticles can cross the endothelium by the paracellular route. These areas, lacking a conventional blood-brain barrier, are located near ventricles (circumventricular organs); they occur in areas of the CNS that respond to substances in the blood. Larger serum proteins and small nanoparticles pass the endothelial barrier in these regions although they may be blocked at the glial barrier. The choroid plexus is one example of an area lacking an endothelial barrier, but in this case, a barrier with tight junctions is present on the choroid plexus epithelium. Even if nanoparticles reach the cerebrospinal fluid, they still cannot directly access the brain parenchyma because of the ependymal/glial barrier. Only a tiny proportion of drugs injected into the blood or ventricles normally reaches the brain parenchyma; for this reason, it is essential when quantitating transport to verify exactly where in the brain the drug/treatment is located [19].

In summary, the treatment of CNS diseases by gene therapy with oligonucleotides has great potential, but gene delivery is the limiting factor. This is particularly true if the nanocarrier must cross the blood-brain barrier to provide widespread distribution in the CNS. Considerable progress has been made with the delivery of smaller therapeutic agents to the CNS by encapsulation within nanoparticles [20], but any type of gene therapy adds an additional problem because the oligonucleotides must subsequently also transfect the target cells. 

## 3. Oligonucleotides for Treatment of CNS Disease

### 3.1. Actions and Modifications of Oligonucleotides

Oligonucleotides (10–50 bp) are too small to encode proteins. They can act at different points in the cell by modulating the production or translation of host mRNA. The ways in which oligonucleotides can act include:Altering splicing of the primary RNA transcript, to include or exclude specific exons.Increasing the degradation of mRNA by inhibiting methylation of the 5′-cap on mRNA.Inhibiting the nuclear export of mRNA by preventing addition of the poly-A tail. This also promotes its breakdown by 3′ exonucleases.Blocking the translation of mRNA by inhibiting ribosome attachment or progress down the mRNA strand.Increasing mRNA’s availability by blocking its interaction with long non-coding RNAs (lncRNA).Forming DNA-RNA hybrids or dsRNA, which are substrates for RNAse H.Promoting the breakdown of mRNA by micro-RNAs (MiRs) or short interfering RNA (siRNA)Acting as guide RNA (gRNA) for gene-editing by CRISPR/Cas9.

Some of these mechanisms are directly due to blocking of active sites on the RNA transcript or mRNA [21], whereas others enlist the cell’s normal machinery to control mRNA production and breakdown [22,23], as summarised in Figure 2 [24]. Antisense oligonucleotides (ASOs), binding to complementary sites on the RNA can sterically block the access by proteins required for processing and translation. Single-stranded DNA is usually preferred for this approach as DNA is more stable than RNA. However, chemical modification of the oligonucleotides (see below) greatly improves the stability of both DNA and RNA [25].

Some cellular processes, including RNA splicing and mRNA breakdown by MiRs and siRNA, specifically require complementary RNA as their substrate. The action of MiRs and siRNA both involve recognition of dsRNA by the cytoplasmic RNA-induced silencing complex (RISC). This complex includes the RNAse argonaute, which selects a guide strand from the MiR or siRNA. The single-stranded RNA binds to complementary sites in the 3′ untranslated region of mRNA and promotes breakdown of the mRNA by removal of the poly-A tail and subsequent loss of the 5′ cap. These processes taking place in the RISC are RNA-specific.

Similarly, gene-editing by CRISPR/Cas9 requires gRNA as a guide to target the complex to the specific point in the genome that is to be edited [26].

The problem of oligonucleotide stability has been partly solved by chemical modification of the nucleotides, in order to increase resistance to nucleases [24]. The first approach was to modify the nucleic acid backbone with phosphorothioate bonds between non-bridging oxygen atoms in the phosphate groups. This change increased the resistance to nucleases, but was not sufficiently effective alone for use in therapeutics. Subsequently, locked nucleic acids (LNAs) with modified ribose/deoxyribose were incorporated into the oligonucleotides either interspersed in the oligonucleotide (mixmer) or at the ends of the oligonucleotide (gapmer). For example, a gapmer with a central targeting segment of modified DNA can be flanked by segments of RNA to act as a substrate for RNAse H. The most effective modifications include substitutions of electronegative groups at the 2′ position of the furanose ring such as the methoxy ethyl (MOE) group. This modification is used in nusinersin, an ASO treatment for spinal muscular atrophy (SMA) [27]. The chemical modification of nucleic acids has substantially improved the resistance of oligonucleotides to nucleases, but it may also reduce the effectiveness of the nucleotide if it must interact with cellular RNA processing machinery.

### 3.2. Level of Therapeutic Oligonucleotides

For treatment of single gene disorders, it may only be necessary to deliver a small number of oligonucleotides to the target cell. For example, correction of a single nuclear gene by gene editing would require a single treatment with a relatively small number of guide oligonucleotides. Conversely, treatment of a condition to reduce cytoplasmic mRNA by binding of ASOs, MiRs or siRNA to the mRNA requires many oligonucleotides to be targeted to individual cells. The same consideration applies for treatments aimed at altering RNA splicing. Moreover, in all of these cases, repeated treatments may be required as the defective nuclear gene continues to produce defective RNA. The problem can partly be overcome if the oligonucleotide is transcribed from a therapeutic transgene integrated into the host cell. However, this in itself requires transfection of the cells with longer gene segments than the therapeutic oligonucleotide, usually with viral vectors.

Many oligonucleotides have been examined in vitro and in animal models of CNS disease for their ability to modulate target gene expression or alleviate disease symptoms. At this time, a small number of oligonucleotide treatments have been approved for use in humans, including treatments for amyotrophic lateral sclerosis (ALS) [28] and spinal muscular atrophy (SMA) [27,29]. Clinical trials in Huntington’s disease have produced disappointing results [30] despite promising results in animal models. Some of the therapeutic oligonucleotides are expressed from adeno-associated viral vectors. All of them are delivered by intrathecal injection.

A number of other oligonucleotides have shown potential in animal models of disease [31,32,33,34,35,36], but they have not yet been translated into treatments for humans. The majority of these treatments have used siRNA or ASOs, but at least one study in macaques has used a MiR to suppress superoxide dismutase [37] and another used an lncRNA in a mouse model of SMA [38].

The selection of the sequence of the oligonucleotide and its nucleotide composition is clearly critically important in determining how the oligonucleotide will act within the cell and whether it will be therapeutically useful. In particular, for conditions such as Huntington’s disease, gene editing must specifically target the mutant allele with an allele-specific oligonucleotide [39]. However, if one considers only crossing of the blood-brain barrier, or uptake by cells, the precise sequence of the oligonucleotide appears to have little effect on how much of it will reach the target cells. The negative charge of the phosphate groups in the backbone of the oligonucleotide is potentially very important in determining uptake by the endothelium or target cells, particularly if the oligonucleotide is on the outside of the nanocarrier.

## 4. Nanocarriers for Oligonucleotides

A large variety of nanoparticle-oligonucleotides have been tested for their ability to transfect cells and for treatment of disease models [14]. Nanoparticles for treatment of CNS diseases have been made in a more restricted range of chemical formulations with sizes ranging from 5–300 nm in diameter [40]. Table 1 summarises the properties of some of the nanocarriers that have been shown to transfect cells in the CNS and/or cross the blood-brain barrier. The size and surface charge are key characteristics of the nanocarriers, affecting their ability to interact with and enter cells. For small nanocarriers, the oligonucleotides are carried on the outside of the nanoparticle and may be covalently or non-covalently attached. Since nucleic acids are strongly negatively charged under physiological conditions, nanocarriers with externally bound oligonucleotides are negatively charged. This has the advantage that it prevents nanoparticle aggregation, but it also affects their interactions with serum proteins and the initial binding to target cells. Strongly-charged nanoparticles attract a corona of serum proteins of the opposite charge, which can neutralise or even reverse the charge and increase the effective size of the nanoparticle [41]. In some cases, the corona proteins or lipoproteins themselves may interact with cell surface receptors (e.g., LRP1) or the endothelial glycocalyx.

### 4.1. Smaller Nanocarriers

For delivery of oligonucleotides to the CNS, various small nanocarriers have been synthesised with cores consisting of gold, silica or iron [42,43,44,45]. Smaller nanocarriers are generally better suited than larger nanoparticles to intravascular delivery as they have greater potential to cross the blood-brain barrier.

Gold glyconanoparticles have a core of 2–4 nm and a surface coat of thiolated sugar residues (glucose or galactose) attached to the core via their sulphur atom [46]. These nanoparticles have approximately 100 gold atoms in the core and 40 sugar residues on the outside [17] and they can cross the blood-brain barrier in vivo (Figure 3) [47]. Other formulations of gold nanoparticles with different reducing agents have also been tested [48]. Thiolated ssDNA oligonucleotides can be attached to the core by an exchange reaction, in which 1–6 oligonucleotides replace the sugar residues. The system can be adapted by hybridisation of cargo oligonucleotides to the original ssDNA attached to the core [43]. The covalently bound oligonucleotides can be released from the core by a further exchange reaction with cytoplasmic glutathione after the nanocarriers have entered the target cell [46]. In the extracellular, non-reducing environment, these nanocarriers are comparatively stable—the rate at which the sulphur-linked cargo is released depends on the redox potential in the intracellular environment.

Mesoporous silica nanoparticles have been regularly used for transport of small drugs, but the pore size is normally too small to accommodate oligonucleotides. (A modification of the synthesis method has allowed larger silica nanoparticles with pores >10 nm to incorporate small oligonucleotides internally.) For gene therapy, silica nanoparticles are usually first coated with cationic molecules and the nucleic acids are non-covalently bound on the outside [44].

Iron oxide nanoparticles with a core of 5–11 nm have been used for imaging in the CNS and to deliver peptides across the blood-brain barrier. The major advantage of these nanoparticles is that they are paramagnetic and can be used for imaging and theranostics. (Magnetic nanoparticles have also been attached to the outside of larger nanocarriers.) External magnetic fields can then be used to induce movement across the brain endothelium [49,50]. However, there have been some concerns about the potential toxicity [51], and partly for this reason, delivery of nucleic acids on these nanoparticles has been for treatment of neuroblastoma where cytotoxic cargoes are used [45] and additional toxicity due to the nanoparticle is not considered problematic.

Cyclodextrins that can form molecular cages containing siRNA have been used in vitro [52].

Liposomes vary greatly in size depending on the number of layers of phospholipid in the shell and on the size and composition of the core, which contains the oligonucleotides (see below).

### 4.2. Larger Nanocarriers

For larger nanocarriers, the cargo is usually incorporated into the core, and is not covalently attached to the nanoparticle. This allows a progressive controlled release, which is more useful for small drugs. More important is that the nucleic acids are better protected from enzymic degradation. Larger carriers can also incorporate targeting antibodies or peptides on the outside, and/or fluorescent or magnetic trackers, to improve localisation to target cells or imaging. Because of their size, these carriers are generally less able to cross the blood-brain barrier, although there are exceptions. Initially, the formulation of these nanoparticles has followed standard practise intended to optimise transfection of the target cell [53]. For these two reasons, treatment in vivo was usually by intrathecal or intraventricular injection. Subsequent modifications have started to address problems associated with delivery across the blood-brain barrier and distribution within the brain. Larger nanocarriers with oligonucleotide cargoes are typically based on polymers, liposomes, lipids or dendrimers. Exosomes have also been developed, although not yet tested for CNS gene delivery.

Polymer nanocarriers include polyethyleneimine (PEI), poly-(lactic coglycolic acid) PLGA, chitosan and collagen [54]. Polyethyleneimine is a cationic polymer that can form nanoparticles (polyplexes) with DNA, where the positively charged PEI condenses with the negatively charged DNA [55,56]. The polyplexes enter the endosomal pathway and are released into the cell following osmotic swelling of the endosome. PLGA nanoparticles can be synthesised in various sizes [57] and have been optimised for transport across brain endothelial cells [58]. However, they have primarily been used for transport of small therapeutic molecules. Chitosan is a linear polysaccharide that can form complexes with DNA [59]. Although chitosan complexes have relatively low transfection efficiency, they are more readily degraded than other polymer complexes so can release their DNA cargo into the cells quite effectively.

Cationic liposomes containing DNA or RNA are small spherical vesicles composed of a hydrophilic core enclosed by a single or multiple phospholipid bilayers [60]. They can be classified according to size and number of bilayers as small unilamellar (10–50 nm), large unilamellar (50–1000 nm) and multilamellar (20–100 nm). They have high transfection efficiency but also a tendency for aggregation under physiological conditions. Some of the limitations of the liposomes can be reduced by chemical modification [61] and they are then effective at protecting the nucleic acid in serum and can cross the blood-brain barrier [62]. 

Protein nanoparticles (human serum albumin) coated with apolipoprotein-A can cross the blood brain barrier and move rapidly within the CNS [63]. However, protein nanocarriers have not generally been used for oligonucleotides.

Lipid nanoparticles in the range of 20–100 nm have potential for transport across the brain endothelium, particularly if suitably modified [64]. siRNA has been incorporated into the nanoparticles to silence the NMDA receptor in neurons following intracerebral injection [65].

### 4.3. Effect of Nanocarrier Size

Nanocarrier size has a major effect on their tissue distribution following intravenous injection [66,67]. In the circulation, nanoparticles <5 nm in diameter are rapidly removed by the kidneys. The fenestrated endothelium of the glomerulus allows relatively free passage of the smallest nanoparticles and they are not reabsorbed in the tubules. Nanoparticles >10 nm diameter are cleared from the circulation by hepatic and splenic mononuclear phagocytes, while those >50 nm can become physically trapped in the open circulation of the splenic red pulp. Attachment of oligonucleotides (20–40 bp) to the outside of 5-nm gold glyconanoparticles increases their size to 7 nm and causes a major shift in their tissue distribution—they are no longer filtered through the kidney but become localised in splenic macrophages and in the liver, mostly hepatocytes [68]. Reducing the losses of nanocarriers <50 nm in the circulation is an important goal since movement across the brain endothelium, the rate of diffusion and transport within the brain, the release of cargo molecules and eventual clearance are all better effected by nanoparticles <50 nm [67,69]. Two advantages of larger nanocarriers are their larger carrying capacity and the slower release of the cargo, although this is more important for small drugs than oligonucleotides.

### 4.4. Targeting of Oligonucleotide Nanocarriers

Due to the very low penetration of the brain by nanocarriers, several methods have been proposed to improve their uptake and/or transport [70]. Surface modification of the nanocarriers can promote uptake by the brain endothelium or the target cells, especially for reducing non-specific uptake in the circulation by hepatic and splenic macrophages. For this purpose, PEGylation is often used to enhance the nanoparticle time in circulation [71]. In addition, attachment of small targeting molecules such as cannabidiol to the outside of the nanocarrier has also been used to improve uptake by the brain endothelium [72].

The brain endothelium has numerous solute carriers for small metabolites, which could, in theory, act as targets for the initial selective binding of circulating nanocarriers to the brain endothelium. The glucose-transporter Glut-1, strongly expressed on the brain endothelium, is a potential target. However, most nanocarriers and oligonucleotides cannot be transported by the physiological transport actions of the solute carriers as the nanoparticles are too large. It is, however, possible to modify an oligonucleotide or a nanoparticle containing oligonucleotides with the substrate (e.g., glucose) so they can engage with the receptors [73,74]. In this case, the substrate improves attachment to the endothelial solute carrier, but does not use its transport function. If chemically-coupled to the nanoparticle, the substrate must be in an appropriate orientation to engage the solute carrier. Alternatively, the solute carrier can be targeted using peptides or other targeting molecules that recognise a site on the carrier separate from the substrate-binding region.

Because of the limitations of the solute carriers, targeting of nanoparticles has focussed on transporters of larger biomolecules, which are taken up by endocytosis and which may then be released at the basal surface of the endothelium [75]. The prime candidates are the transferrin receptor (TfR, CD71), the low-density lipoprotein receptor LRP1 (CD93) and the insulin receptor [76]. The substrates are internalised by the brain endothelium and a variable proportion may be used by the cell or transported across the basal membrane. It is debatable whether any insulin enters the brain by this route, but there is good evidence for at least some transcytosis of transferrin and LDL. Although the receptors are strongly expressed on the brain endothelium, they are also present in many other cell types—targeting can improve CNS uptake, but the improvement is relative to the low initial baseline.

TfR1 is involved in uptake of the iron-transport protein transferrin, and is expressed particularly on dividing cells. It recycles between the apical membrane of the brain endothelium and endosomes where the transferrin is uncoupled. A smaller proportion of the receptors transfer to the basal membrane. The receptor has been targeted by antibodies, antibody fragments and peptides [77,78]. Interestingly, antibodies of moderate affinity are more effective at mediating transcytosis of cargo molecules than high affinity antibodies since they can release at the basal membrane and they do not cause the TfR to be diverted for breakdown in endothelial lysosomes [79]. One limitation of antibodies for targeting is their relatively large size—a single antibody (160 kDa) is larger than a small nanocarrier (15–30 kDa), meaning attachment of antibodies to smaller nanocarriers substantially increases their size and may also alter their charge.

The LDL receptor LRP1 transports cholesterol and lipids for neurons and glia, but it is also involved in removal of amyloid-β from the brain. As such, it shuttles between the apical and basal membranes of the endothelium and can bind nanoparticles with surface apoE [80]. Cholesterol has also been directly conjugated to nucleic acids to enhance transport across the brain endothelium [81], and this has been proposed as a simple method to promote delivery of ASOs to the brain following intravenous injection.

Both TfR1 and LRP1 have been targeted for transport of nanoparticles into the brain by attaching receptor-binding polypeptides to the outer surface of the nanoparticles [82]. It is important when using these systems for transporting nanocarriers that they do not significantly interfere with transport of the physiological substrate. This could occur by direct blocking of the substrate binding site or by promoting breakdown of the receptor. In practice, the serum concentrations of the natural substrates (transferrin or LDLs) and their affinity for their receptors is usually much greater than the effective concentration of nanocarriers, so interference with physiological transport systems is not problematic.

Finally, a number of physical methods have been used to enhance nanoparticle delivery in vivo, including a focussed ultrasound to increase the transport of gold nanoparticles [83] and magnetic fields for iron or magnetite-coated nanoparticles [49,50].

## 5. Tissue Distribution of Oligonucleotide Nanocarriers

Delivery of large therapeutic biomolecules to the CNS is difficult. For proteins and peptides injected intravenously, the concentration in the brain intercellular space is usually much less than 0.5% of the level in serum, but is dependent on the size, shape and charge of the biomolecule [66]. Similar considerations apply to nanocarriers; any level in the brain greater than 1% of the serum concentration has been considered as significant delivery. Hence, nanocarriers have mostly been delivered by intrathecal or intraventricular injection. In this context, the anatomical difference between animal models of disease and humans is very important. Following intrathecal injection into a mouse brain, as all areas of the brain lie within a few millimetres of the injection site, a local injection may spread through a considerable proportion of the brain. By comparison, in humans, a local injection will mostly remain within the injected subregion of the brain. This may be desirable if the intention is to treat a local area or a well-localised tumour. However, for many genetic diseases, it is desirable to treat all areas of the brain.

### 5.1. Route of Administration

Studies on intraventricular injection of RNA in animals have shown that it spreads over a surprisingly large distance into the tissue. In humans, the success of treatments for ALS and SMA as well as observation of post-mortem tissue implies that a significant dose of oligonucleotide can reach cells in the spinal cord, presumably by diffusion of CSF through the subarachnoid space and central canal and transfer across ependymal cells [28]. However, the lower rate of success in human brain diseases may reflect the difficulty in delivering the oligonucleotides throughout the brain, more than problems with uptake by the target cells [11].

A small number of studies have used the olfactory route to deliver oligonucleotides [84] or nanoparticles with oligonucleotides [85,86,87] to the brain. The siRNA was confined to the olfactory bulb, whereas the formulation on chitosan nanoparticles produced a reduction of gene expression (HTT) in the hippocampus, striatum and cortex. Intranasal delivery is an attractive option because it is non-invasive and can be repeated, but it is likely to translate less well to humans because of the small size of the olfactory bulbs in humans and their distance from potential target areas.

Because of these anatomic factors, it would be very desirable to deliver nanocarriers across the blood-brain barrier when treating conditions in humans where large areas of the brain are affected. It is estimated that all cells of the CNS lie within 100 µm of a capillary. Consequently, if the blood-brain barrier can be overcome, the diffusion distance for nanocarriers within the brain is relatively small. It might be thought that the tortuous intercellular spaces within the brain would still make the diffusion pathway for nanoparticles much longer than the direct distance from the capillary to the target cell. However, with both lipid nanoparticles and gold nanoparticles, it appears that nanoparticles that have crossed the blood-brain barrier can move relatively quickly within cells of the brain parenchyma (up to 5 µm per minute), possibly using intracellular transport systems [47,80].

**Table 1 ijms-23-00760-t001:** Oligonucleotide nanocarriers for delivery to CNS.

Nanocarrier	Cargo	Notes	Ref
Glucose/galactose-coated 2 nm gold core	Thiol-bound ssDNA or dsDNA 20–40 bp	7–8 nm	[43]
Polymer-modified mesoporous silica	Internal, ASOs	70–200 nm	[44]
PEI or amine coated 15 nm iron-oxide core (SPIO)	External, electrostatic-bound ASO	Superparamagnetic 50–60 nm	[45]
Aminated, cationic cyclodextrin	Trapped siRNA	160–180 nm, peptide-targeted	[52]
Ca phosphate core, phospholipid shell with bound PEG	ssDNA ASO in core	30–60 nm	[53]
Polyethylene imine (PEI) modified with PEG	Trapped ssDNA ASO	90–160 nm, insulin/transferrin targeted	[55]
Linear PEI/PEG conjugate	siRNA	Fibrillar micelles formed around RNA	[56]
Bioreducible lipids modified with cholesterol/DOPE/PEG	ASO in core	150–500 nm with ASO	[61]
Tri-poly phosphate-modified chitosan/PEG conjugate	Encapsulated ssDNA ASO	170 nm nanoparticle + TfR antibody = 784 nm	[59]
Liposome—cationic lipid (DOTAP) and cholesterol ± PEG	Encapsulated siRNA	Conjugated peptide targeting AcChR	[62]
Liposome—cationic lipid mixture	siRNA duplex	50–60 nm	[65]
Chitosan	siRNA	103–205 nm	[87]
Peptide-tagged, chitosan/PEG	siRNA-biotin	5–10 nm	[86]
Lipochitoplex—chitosan core/liposome shell	DNA in chitosan core	Chitosan core 65 nm Lipochitoplex 99 nm	[88]
Polyion complex micelle + modified poly-L-lysine and PEG	ASO in core	45 nm targeting GLUT1	[73]
Cationic lipid mixtures	Ribonucleoprotein—gRNA/DNA, Cas9	<200 nm, dependent on formulation	[26]

### 5.2. Subcellular Localisation of Nanocarriers and Oligonucleotides

When crossing the brain endothelium, nanocarriers can move via the transmembrane/cytosolic route or by vesicular transcytosis (Figure 4). Those entering the cytosol are exposed to the reducing conditions normally found in the cytoplasm. In contrast, those moving by vesicular transcytosis may be subject to low pH in the endosomal compartments. In either case, the rate of transit across the endothelial cell is usually less than 30 min, which is not long enough to cause significant loss of the cargo. For example, gold glyconanoparticles retain a thiol-bound cargo for several hours in the conditions normally found in the endothelial cytosol (1–10 mM glutathione) and are fully stable down to pH 5 [46].

The intracellular route taken by a nanocarrier depends on its size, charge and surface ligands. Electron microscope studies imply that some small nanoparticles and larger lipophilic nanoparticles are taken up directly into the cytoplasm—no membrane is visible around the nanoparticle. More often for nanoparticles in the range of 10–100 nm, they are initially taken up into membrane-bound vesicles, usually caveolae.

A potential problem during transcytosis is loss of the nanocarrier and cargo by diversion of endosomes to the endothelial lysosomes [11]. The extent that this occurs is little researched although it appears to vary considerably depending on how the nanoparticle has interacted with any cell surface molecules—it appears that high-affinity binding and/or cross-linking of cell surface receptors by the nanocarrier is more likely to cause the receptor and its bound nanoparticle to enter the lysosomal pathway.

Ultimately, transported oligonucleotides must enter the cytoplasm of the target cells. As noted earlier, nanocarriers used for direct intrathecal injection have usually been optimised for their transfection efficiency. However, nanocarriers optimised for transport across the brain endothelium by vesicular transport also appear to enter endosomes in target cells, and they must escape into the cytoplasm before they can have a biological effect.

## 6. Conclusions

Nanocarriers offer several advantages for the delivery of therapeutic oligonucleotides to the CNS by protecting the nucleic acid, increasing transport across the blood-brain barrier and improving transfection of the target cells. A great variety of nanocarriers have been used for this purpose, varying in chemical formulation, size, charge and carrying capacity. No single nanocarrier has the best characteristics for transport into the brain and transfection efficiency. Moreover, any nanocarrier injected intravenously has to first avoid removal via the kidney or clearance by mononuclear phagocytes in the spleen or liver. For this reason, all oligonucleotide treatments approved for use in humans to date are delivered by intrathecal injection. Several nanocarriers with oligonucleotide cargoes have shown effective CNS gene modulation in experimental animal models of disease. This suggests that intravenous delivery of nanocarriers may also be possible in humans. However, the anatomical differences between humans and rodents mean that gene delivery across the blood-brain barrier is both more desirable in humans than intrathecal injection and also considerably more difficult.

## Figures and Tables

**Figure 1 ijms-23-00760-f001:**
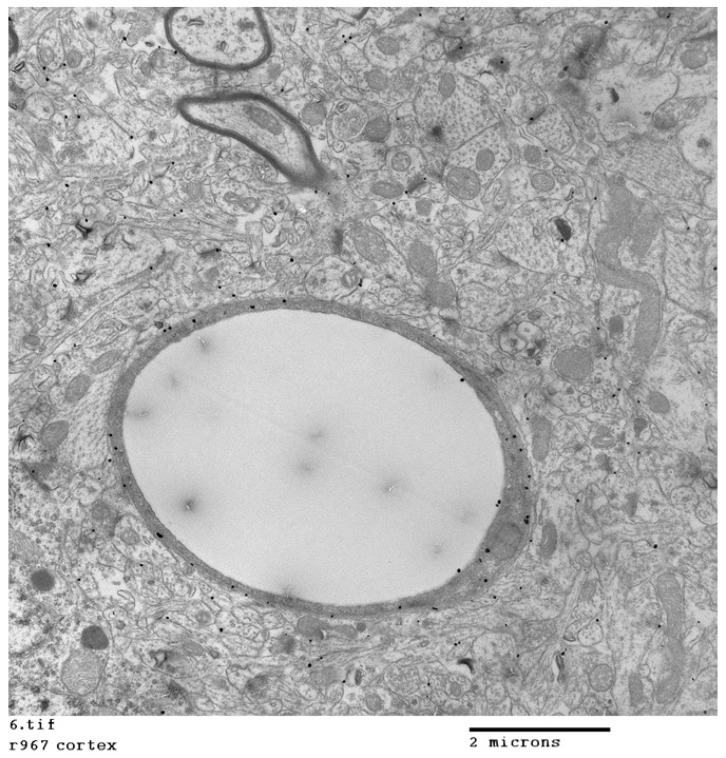
Transfer of 5-nm pegamine-coated gold nanoparticles across the rat brain endothelium. A silver-enhanced TEM image of cortex 10 min after intra-carotid infusion shows nanoparticles (black electron-dense dots) located in the endothelium of a capillary, in astrocyte foot processes adjoining the vessel and in neurons up to 4 µm from the vessel. Image is courtesy of Dr. Radka Gromnicova.

**Figure 2 ijms-23-00760-f002:**
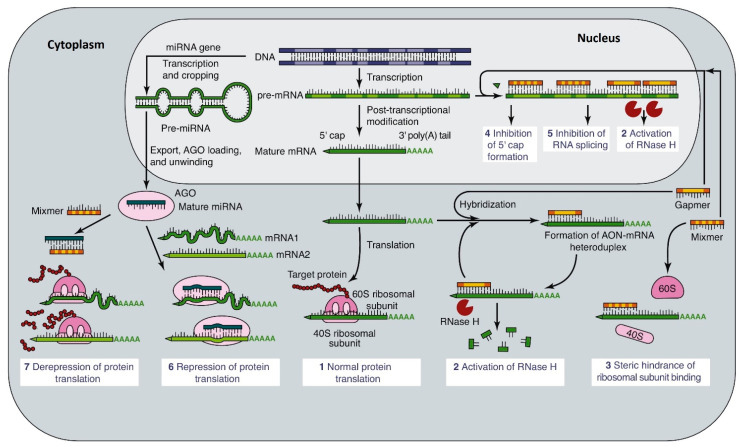
Summary of the ways in which antisense oligonucleotides can modulate normal gene expression (1). Binding of ASOs to mRNA produces double-stranded segments susceptible to cytoplasmic RNAse-H (2). Binding of ASOs to the 5′ or translated segments of mRNA interferes with ribosomal assembly or translation (3). In the nucleus, ASOs binding to the primary transcript can inhibit cap formation (4), polyadenylation or splicing (5), with the potential to act as a substrate for RNAse H. MiRs processed by the RISC complex (or directly transfected siRNA) can inhibit translation by binding to the mRNA (6), or siRNA can derepress translation by interfering with the binding of the mRNA with lncRNAs that inhibit mRNA availability (7). Adapted from Figure 1 in Hagedorn et al. [24].

**Figure 3 ijms-23-00760-f003:**
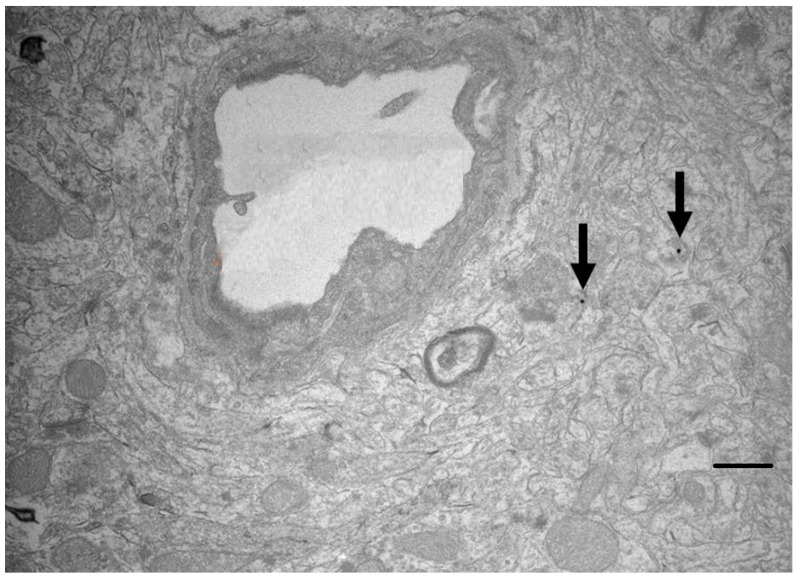
TEM silver-enhanced image showing transport of 7-nm gold glyconanoparticles carrying 40 bp dsDNA oligonucleotide (arrows) into the cortex of the rat brain 10 min after infusion into the carotid artery. Scale bar = 500 nm. Image is courtesy of Dr. Nayab Fatima.

**Figure 4 ijms-23-00760-f004:**
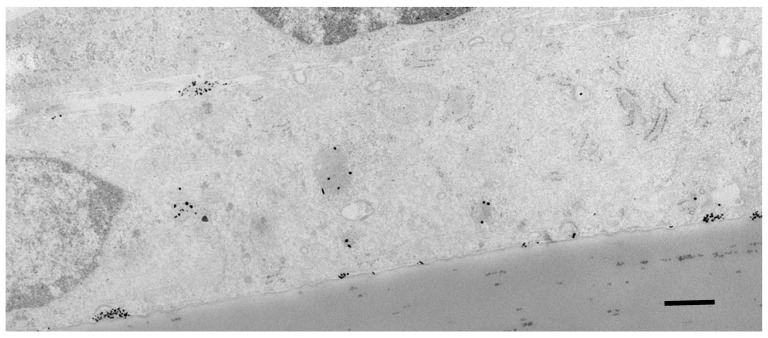
TEM silver-enhanced image showing transport of 7-nm gold glyconanoparticles carrying 40 bp dsDNA oligonucleotides across the human brain endothelial cell line hCMEC/D3 in vitro. Nanocarriers (electron-dense black dots) are transported from the apical surface of the cell (upper) to the basal surface (lower) both in vesicles and by cytosolic transfer. Nanocarriers can be seen in clusters released from vesicles at the basal membrane. Image is courtesy of Dr. Nayab Fatima. Scale bar = 100 nm.

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
