# Peer review of "Nanocarriers for Delivery of Oligonucleotides to the CNS"

_ijms, 2022, doi:10.3390/ijms23020760_

Round 1

Reviewer 1 Report

I think this is a well written and comprehensive review of the use of nanoparticle formulations for the delivery of oligonucleotides as therapeutics.  My only criticisms are as follows:

I found the same typo in multiple places and this may be the result of an autocorrection.  On lines 133, 385, 390 you used the unit µM (unit of concentration) instead of µm (unit of distance).

on lines 179-181 you discuss chemical modifications of oligonucleotides and mention the use of locked nucleic acids (LNAs) as a sugar modification on the molecule.  This statement made it seem that there were no other sugar modifications used in oligonucleotides when in fact there are multiple modifications.  I think it would be better in this section to talk about sugar modifications and provide a couple of examples such as LNAs and methoxyethyl (MOE, the modifications in Spinraza).

Other than those two criticisms I think the authors did an excellent job reviewing the complex issues around the formulation of oligonucleotides using nanoparticles and other modalities.

Bravo!

Author Response

The typographical errors relating to units of length have been corrected.

An addition (with a reference) has been made to the text on locked nucleic acids, and nusinersin (Spinraza) is given as an example of an MOE modified anti-sense oligonucleotide. (line 180 -)

Reviewer 2 Report

This paper is a review of interest to many people interested in learning about the use of oligonucleotides as potential therapeutic tools. The review is ambitious in topics, related to both basic aspects of limiting anatomical barriers for the passage of nanocarriers and the mechanisms of oligonucleotides, as well as others of a more applied nature related to the size and targeting of nanoparticles of potential interest in the pathologies of central nervous system .

The breadth of topics addressed limits the depth in which they are analysed. But, the most relevant questions are considered. Thus, the review retains its interest for both readers specialized or not, in the area of oligonucleotides with therapeutic interest.

However, there are some issues that authors should include or modify in the manuscript:

  1. Line 140: the title of this section (“3. Oligonucleotides for treatment of CNS disesases”) has a single subsection (“3.1 Leve lof therapeutic oligonucleoties”) and therefore it is not necessary to list it. It is enough that it is highlighted in bold. However, since the introductory text to subsection 3.1 is mechanistic in nature, I suggest that it can be considered as section 3.1 by referring in the title to its content on the mechanism of action and modifications of oligonucleotides. In this way the current section 3.1 will be 3.2
  2. Line 232: In the nanocarriers section there is no mention of small liposomes. Do you consider it appropriate? Please consider this point.
  3. Lines 385-390: the values presented (100 µM and 5 µM) are expressed in molar units. If it is intended to be expressed in units of length (and not concentration), they must be modified by 100µm and 5µm.
  4. Line 437: the following review article should be included in the References section and duly cited in the manuscript: Zohou Y, et al Crossing the blood-brain barrier with nanoparticles. Journal Controlled Release 270 (2018) 290-303

Author Response

1. The organisation of material in section 3 now includes an additional subsection and heading.

2. Liposomes can be any size from 10 -500nm. They are reviewed here under the heading ‘larger nanocarriers’ , but since they also fall into the category of ‘smaller nanocarriers’ , an extra sentence with forward reference has been added to the section on smaller nanocarriers (line 264 -).

3. The units have been corrected.

4. The review by Zhou et al 2018 has been added to the reference list [20] and is cited in the text in relation to nanocarriers. The following references have been renumbered.